# Domestic dogs maintain clinical, nutritional, and hematological health outcomes when fed a commercial plant-based diet for a year

**Annika Linde**[1], **Maureen Lahiff**[2], **Adam Krantz**[1], **Nathan Sharp**[1], **Theros T. Ng**[1], **Tonatiuh Melgarejo**[1]*

**1** Western University of Health Sciences, College of Veterinary Medicine, Pomona, California, United States of America, **2** University of California Berkeley, School of Public Health, Berkeley, California, United States of America

☯ These authors contributed equally to this work.

* tmelgarejo@westernu.edu

**Data Availability Statement:** All relevant data are within the manuscript and its Supporting Information files.

## Abstract

Domestic dogs can maintain health on complete and well-balanced canine plant-based nutrition (K9PBN). Novel insight on health outcomes in dogs consuming K9PBN is of relevance to veterinary professionals and consumers given a growing interest in non-traditional dog foods with perceived health benefits, while considering potential safety concerns. We aimed to investigate nutritional equivalence by measuring clinical health outcomes in adult dogs fed K9PBN over twelve months compared to a meat-based diet at baseline. We enrolled fifteen clinically healthy adult dogs living in households in Los Angeles County, California in a prospective cohort study and evaluated clinical, hematological, and nutritional parameters in dogs at 0, 6, and 12 months, including complete blood count (CBC), blood chemistry, cardiac biomarkers, plasma amino acids, and serum vitamin concentrations. The study found that clinically healthy, client-owned, adult dogs maintain health, based on physical exams, complete blood count, serum chemistry, plasma amino acids, serum vitamins, and cardiac biomarkers combined with client-reported observations, when fed commercial K9PBN over a twelve-month period. This study is the most comprehensive and longest known K9PBN investigation to date and provides clinically relevant evidence-based nutrition data and new knowledge on outcomes in clinically healthy dogs who thrive without consumption of animal-derived ingredients. These results also provide a valuable foundation for the future study of K9PBN as a potential nutritional intervention for clinically relevant pathologies in canine medicine. Lastly, it is of major relevance to One Health paradigms since ingredients produced independent of industrial food animal production are both more sustainable and help to circumvent ethical dilemmas for maintenance of health in domestic dogs.

**Funding:** The study was made possible with funding from the Plant-Based Dog Food Health Study Initiative in Los Angeles, California. The funders had no role in study design, data collection and analysis, decision to publish, or preparation of the manuscript.

**Competing interests:** The authors have declared that no competing interests exist.

## Introduction

Health relies on species-appropriate nutrition that aligns with different life stage requirements to optimize immune function while resisting disease-promoting stressors and dangers [1] and also remaining cognizant that determinants of health span across five domains, including genetics, behaviors, environmental exposures, health care, and social circumstances [2]. In comparison to the carnivorous wolf (*Canis lupus*), the domestic dog (*Canis familiaris*) is an omnivore species based on phylogenetic context and key anatomical-physiological traits [3]. Evolutionary adaptations have resulted in a digestive system that enables dogs to maintain health on nutritionally complete omnivorous diets, including those free of animal ingredients —demonstrating dogs have nutrient, not ingredient, requirements [4]. Compared to the digestive capabilities of its ancestors, the domestic dog has developed adaptations in carbohydrate metabolism that enable more effective starch digestion [5]. This important adaptation also enabled dogs to continue their co-existence with early humans during the agricultural revolution. The result is reflected in modern dogs, where even canine endurance athletes are able to maintain a high level of performance without animal-derived products in their diet [6]. Increasing awareness of industrial food animal production's central role in driving accelerated anthropogenic climate change, environmental degradation, disease patterns as well as social injustices, has impacted consumer behaviors and fueled a growing interest in products developed without animal-sourced ingredients [4, 7]. Changing societal trends have created an increasing demand for non-conventional dog foods, which has also triggered questions about the nutritional adequacy of such diets and their impact on cardiac and other health outcomes in canine consumers [8–13]. While short-term studies have supported the adequacy of K9PBN [4, 6, 14], the need for long-term studies has remained. Bridging this knowledge gap served as the rationale driving this study. The study aim was to assess nutritional equivalence by measuring long-term health outcomes through evaluation of clinical, hematological, and nutritional parameters in clinically healthy adult dogs fed a well-balanced, high-quality, commercial plant-based diet with pea protein as a main ingredient over twelve months as compared to commercial meat-based diets at baseline. For the purpose of this study, the term "meat-based" is defined as a product with animal-derived protein sources regardless of if the product contains plant-derived ingredients. The term "plant-based" is defined as a product with plant-derived protein sources without animal-derived protein sources. The term "high quality" refers to alignment with AAFCO (Association of American Feed Control Officials) nutrient profiles (adult maintenance). The overall hypothesis fueling the study is that parameters of clinical health are maintained within normal limits after a year of consuming K9PBN. We showed that clinically healthy, adult, dogs can maintain health over a longer period of time when using a complete K9PBN approach.

## Materials and methods

### Study design

A within-subject design with repeated measures was used for quarterly assessment of health status in fifteen adult dogs fed commercial K9PBN between January 2020 and January 2021. All dogs were fed different commercial meat-based diets for at least one year prior to enrollment, allowing individuals to serve as their own controls and strengthen the precision of the study. Clinically healthy canine participants were prospectively identified by emailing all students, staff, and faculty at Western University of Health Sciences (WesternU) in California. The study was approved by the WesternU Institutional Animal Care and Use Committee (R18IACUC014) with client consent obtained for all participating dogs prior to enrollment in

the study. Incentives included a one-year supply of the study diet plus plant-based treats (V-Dog Kind Kibble and Wiggle Biscuits) and a gift card upon study completion. All dogs were examined at the WesternU Pet Health Center (PHC), and subject to quarterly physical examinations by a licensed veterinarian, including recording of body weight and body condition score (using a 9-point scale: underweight (1–3), ideal (4–5), overweight (6–9)) [15], and sample collection (blood, urine, feces) at 0, 6, and 12 months. The veterinarian evaluating the dogs was not blinded. While the study was designed to examine dogs every three months, we expected analyses of samples from the baseline, intermediate, and endpoint of the study would be adequate to fulfill the objectives. Additionally, we conducted monthly interviews with clients during which we also inquired if food and treat amounts were given as instructed (following the manufacturer's feeding recommendations). Client telephone questionnaire (non-validated) included the following questions: 1) How is your dog's appetite? Any food consumption changes?, 2) Any water consumption changes (frequency)? Any change in water intake volume?, 3) Any changes in your dog's defecation pattern? Any changes in fecal consistency, number of bowel movements? Any sign of pain or difficulty defecating?, 4) Any changes in your dog's urination pattern? Any changes in urine color or odor? Any sign of pain or difficulty urinating?, 5) Any changes in your dog's body weight (note if scale versus visual)?, 6) Any changes in your dog's physical appearance? Appearance of coat? Muscle mass?, 7) Any unusual changes in your dog's behavior. Is your dog bright, alert, responsive?, 8) Any concerns you want to share with us about your dog in reference to this feeding trial?

## Nutrient composition of the study diet

V-Dog Kind Kibble (V-Dog, San Francisco, CA) was selected because it is formulated to meet AAFCO nutrient profiles for adult maintenance, and it is the longest-standing manufacturer in the K9PBN category in the US market (since 2005). The nutrient composition of the K9PBN study diet (crude protein, amino acids, crude fat, omega-6/omega-3 fatty acid ratio, minerals, vitamins and others) and glyphosate (a widely used herbicide) levels was analyzed by Eurofins Scientific Inc. Nutrition Analysis Center (Des Moines, IA) and compared to AAFCO nutrient profiles for adult maintenance (dogs) [16, 17] and published literature on allowable daily intake (ADI) of glyphosate for humans [18]. The sample for nutrient composition analyses and the study diet fed to the dogs originated from a single batch.

## Hematology, blood chemistry, and urinalysis

Blood was collected via cephalic, jugular, or lateral saphenous venipuncture for CBC and blood chemistry, and nutrient biomarker analyses. Venipuncture was performed in dogs at each time point to collect 8 ml of blood divided into 6 ml in red cap-yellow ring (serum clot activator) tubes, 1 ml purple (EDTA), and 1 ml green (lithium heparin) top tubes, which was used for chemistry, hematology, and nutrient biomarker analyses. Urine (2 ml) was collected (ultrasound-guided cystocentesis) into white top tubes (no additive) for urinalysis. A fresh fecal sample was collected from each dog by clients prior to each exam, and stored at -80˚C for future analysis. In-house diagnostic evaluations were performed at the WesternU PHC using equipment from IDEXX Laboratories (Westbrook, ME), including a ProCyte Dx Hematology Analyzer (CBC), Catalyst One Chemistry Analyzer (serum chemistry and T4), and VetLab UA Analyzer & SediVue Dx (urinalysis). Batches of serum/plasma samples were also stored at -80˚C for future analyses.

Cardiac troponin I (cTnI) analysis was performed by shipping serum sample aliquots to Texas A&M University (TAMU GI Lab, College Station, TX), which uses a high-sensitivity ADVIA Centaur cTnI assay from Siemens Healthineers (Siemens Medical Solutions USA,

Inc., Malvern, PA) that has been validated for use in dogs [19]. NT-proBNP analysis was performed using the Bionote in-clinic, fluorescence immunoassay diagnostic analyzer Vcheck V200 and a canine NT-proBNP assay (Bionote USA, Big Lake, MN), which has also been validated for clinical use [20].

## Nutrient biomarker analyses

The nutrient biomarker analyses (plasma amino acids, serum L-carnitine, serum liposoluble vitamins, serum water-soluble vitamins) were completed by third-party laboratories at the University of California Davis (UCD, Amino Acid Laboratory, Davis CA), Eurofins Craft Technologies (Wilson, NC), Michigan State University (MSU, Veterinary Diagnostic Laboratory, Lansing MI), and Texas A&M University (TAMU, GI Laboratory, College Station TX), respectively. All samples were shipped to each laboratory in a single batch overnight with iced gel packs.

Plasma was separated from 1 ml whole blood collected into a green top tube (lithium heparin), and submitted to UCD for Plasma Amino Acid (AA) Analysis. Plasma AA concentrations were measured by the Amino Acid Laboratory at UCD by means of an automated high-performance liquid chromatography AA analyzer as previously described [21].

Serum was separated from 6 ml whole blood collected into red cap-yellow ring (CAT Serum Sep Clot Activator) tubes, and sub-divided into 0.5 ml aliquots. One aliquot was sent for L-carnitine analysis. Serum L-carnitine concentration was measured by Eurofins Craft Technologies (Wilson, NC) according to existing laboratory procedures. Samples were diluted and filtered through 10 kDa Amicon Ultra-0.5 mL (Cat. No. UFC501024, EMD Millipore) before analysis using an L-carnitine Assay Kit (Cat. No. AB83392, ABCAM). Colorimetric reactions were measured using a VersaMax tunable microplate reader, following assay kit procedure.

Additionally, 0.5 ml serum sample aliquots were sent to MSU and TAMU for vitamin analyses. Analyses of liposoluble vitamin 25-Hydroxyvitamin D, vitamin A and vitamin E (alpha tocopherol) were performed by the Veterinary Diagnostic Laboratory at MSU (Lansing, MI). Water-soluble vitamin $B_{12}$ (cobalamin), and vitamin $B_9$ (folate) were analyzed by the GI laboratory at TAMU (College Station, TX) according to internal laboratory protocols.

## Statistical analysis

A non-parametric Wilcoxon matched-pairs signed rank test was used to compare health profiles (clinical, nutritional, and hematological parameters) of dogs at baseline versus endpoint (0 and 12 months), while a Friedman test was used to compare health profiles at baseline, intermediate and endpoint (0, 6 and 12 months). Non-parametric testing was chosen in light of the sample size. Unlike t-tests with small sample sizes, these non-parametric distributions do not assume normal distributions. Still, their power to detect differences is almost as high. Data are presented as median and either $Q_1$ to $Q_3$ (first quartile—third quartile), or (minimum—maximum), which was determined based on the format of available reference values for each of the measured parameters. P-values $< .05$ were considered statistically significant. Data analysis was performed using GraphPad Prism 9 version 9.4.1 for macOS (GraphPad Software, San Diego, CA).

To determine the minimal required sample size, an a priori power analysis was carried out utilizing the effect size from a recent study on health outcomes in dogs fed a plant-based diet [14]. Plasma taurine concentration from that study was used in the power analysis given the role of this amino acid in some dogs with taurine-deficiency dilated cardiomyopathy [22]. The required sample size using the effect size (d = 1.4) of the plasma taurine concentrations was

calculated to be 7 to achieve 0.80 power (1 – β error probability with an α error probability of 0.05 using the Wilcoxon signed-rank with one sample case test). Since we included 15 dogs in this study, which used a within-subject design with repeated measures that allowed subjects to serve as their own controls, our sample size is large enough to detect the size difference we sought with an actual power of 0.83. Sample size and power calculations were completed using G*Power version 3.1.9.7 [23].

## Results

### Dogs

Dogs enrolled in the study had a median age of 4 years (range 1–11 yrs.) and 53% were male (8 of 15). All dogs were altered (spayed/neutered) except one intact male. Breeds represented (one dog per breed) included, American pit bull terrier, boxer, English bulldog, French bull-dog, springer spaniel, Labrador retriever, while the remaining 67% were mixed breed (5, 3, and 1 derived from terrier, herding, and sporting group breeds, respectively). The median (minimum–maximum) body weight was 20.2 kg (5.4–41.4 kg) at baseline, 20.9 kg (5.4–40.7 kg) at 6 months (intermediate), and 20.6 kg (6.2–41.8 kg) at endpoint, showing there was no statistically significant change in body weight between the three timepoints (p = 0.08). Two dogs (1 English bull dog, 1 German shepherd mix) were obese upon enrollment, and their body condition score decreased from 8 to 7 (scale of 1–9). One dog (shepherd mix) was overweight upon enrollment and BCS decreased from 7 to 6 at endpoint (**Fig 1**). The remaining 80% (12 of 15) of the dogs maintained a normal body condition score of 5 (scale of 1–9). The quarterly physical exam findings combined with monthly client reported histories confirmed that all dogs remained clinically healthy while fed K9PBN over twelve months. No adverse effects were reported for any of the dogs in response to the eight questions included in the client telephone questionnaire.

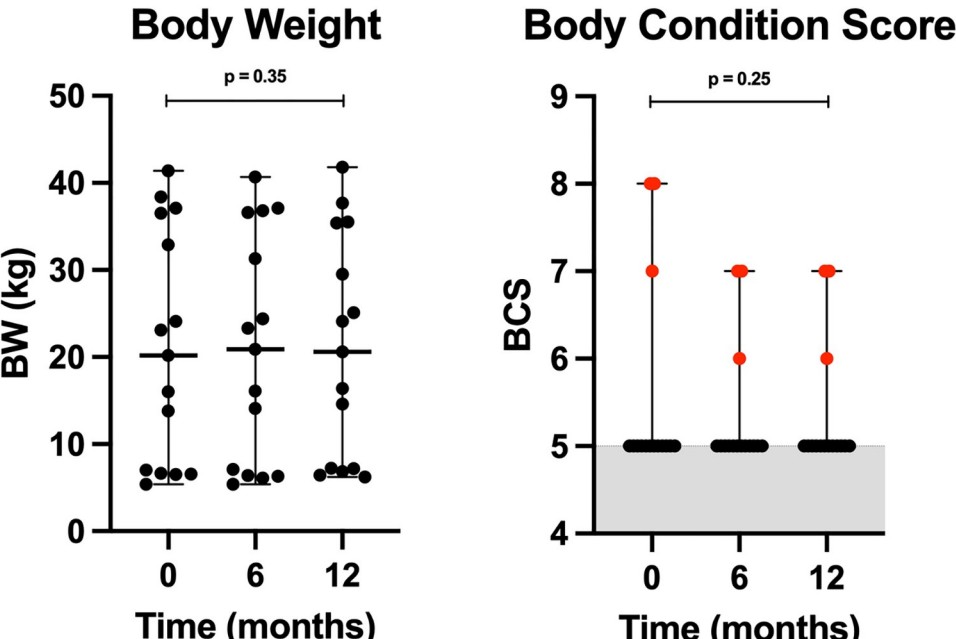

**Fig 1. Scatter plots of body weight and body condition score (BCS) in dogs at 0, 6, and 12 months.** Black and red data points represent dogs with normal and increased BCS, respectively. The grey-shaded area represents normal range for BCS (4–5).

## Nutrient composition of the study diet

Analysis of the study diet (Kind Kibble, V-Dog, San Francisco, CA) confirmed nutrient levels adherent to AAFCO nutrient profiles for adult canine maintenance (**Table 1**). Additionally, the glyphosate level in the kibble (0.18 mg/kg) measured substantially below the ADI for humans [18].

## Complete blood count, blood chemistry, and urinalysis

Comparative analysis of blood work and urinalysis showed no clinically significant changes between baseline, intermediate, and endpoint values. CBC and blood chemistry values remained within normal reference intervals (**S1 Table**), as provided by IDEXX Laboratories (Westbrook, ME), or DiBartola 2012 in the case of osmolality [24].

The urinalysis results showed no statistically significant differences between baseline, intermediate, and endpoint values (**S2 Table**). Normal values [25] assume specific gravity is evaluated in light of hydration status, blood work, and history. We identified crystals in the urine from 60% of the dogs (9 of 15) with no identifiable pattern to the changes. Crystals were identified at different time points, including 3 dogs at baseline (1 with unclassified crystals; 1 with struvite & unclassified crystals; 1 with calcium oxalate dihydrate, struvite & unclassified crystals), 3 different dogs at 6 months (2 with struvite & unclassified crystals; 1 with bilirubin & unclassified crystals) and 3 different dogs at end-point (1 with calcium oxalate dihydrate; 2 with struvite & unclassified crystals). One dog had crystals in the urine at both 0 and 12 months (struvite & unclassified crystals), while no crystals were identified at 6 months. Potential crystal formation is associated with changes in urine pH [26, 27]. All dogs were clinically healthy and hydration status remained within normal.

## Serum cardiac biomarker concentrations

Serum cTnI concentrations showed no statistically significant differences between baseline, intermediate and endpoint values (p = 0.08), or between baseline and endpoint values alone (p = 0.08). An overall decrease was observed in median (minimum–maximum) values between baseline 62.4 pg/ml (13.2–352.6 pg/ml), intermediate 64.61 pg/ml (11.0–185.5 pg/ml), and endpoint 29.6 pg/ml (6.3–208.9 pg/ml). cTnI levels measured within normal (0–49 pg/ml) in 7, 8, and 10 dogs at 0, 6, and 12 months, respectively. cTnI levels were equivocal (50–135 pg/ml) in 5, 3, and 4 dogs at 0, 6, and 12 months, respectively. Levels were consistent with active myocardial injury ($\geq$ 136 pg/ml) in 3, 4, and 1 dog at 0, 6, and 12 months, respectively (**Fig 2**). All study participants were clinically healthy adult dogs with no clinical signs of heart disease.

Serum N-terminal pro-brain natriuretic peptide (NT-proBNP) concentrations showed no statistically significant difference between baseline, intermediate, and endpoint values (p = 0.25). A statistically significant difference was found between baseline and endpoint (p = 0.04). A decrease was observed in median (minimum–maximum) values between baseline 586.7 pmol/L (500.0–1,146 pmol/L), intermediate 555.5 pmol/L (500.0–698.5 pmol/L), and endpoint 544.6 pmol/L (500.0–722.0 pmol/L). NT-proBNP levels measured within normal (< 900 pmol/L) in 12, 15, and 15 dogs at 0, 6, and 12 months, respectively. Levels were 'suspected' (900–1,800 pmol/L) in three dogs at baseline (**Fig 1**). None were abnormal (>1,800 pmol/L) at any of the timepoints.

## Plasma amino acids, including L-taurine, and serum L-carnitine concentrations

The interquartile intervals (middle 50% of data delineated by the first and third quartiles) of all essential amino acid (AA) levels (**Fig 3**) measured within or above the reference intervals ($Q_1$ –

**Table 1. Nutrient profile of the study diet analyzed by Eurofins converted to units per 1,000 kcal ME and compared to the AAFCO dog food nutrient profiles based on caloric content for adult maintenance (AAFCO 2019 Official Publication, p.156-7).** ND: not determined. ω-6/ω-3 FA ratio = omega-6/omega-3 fatty acid ratio = (LA + AA):(ALA + EPA + DHA) = (linoleic + arachidonic):(alpha-linoleic + eicosapentaenoic + docosahexaenoic) acid ratio.

| Nutrient | Units per 1000 kcal ME | Study Diet (Eurofins Analysis) | AAFCO K9 Adult Maintenance | |
|---|---|---|---|---|
| | | | Minimum | Maximum |
| *Crude protein* | g | 75.30 | 45.0 | ND |
| Arginine | g | 5.39 | 1.28 | ND |
| Histidine | g | 1.74 | 0.48 | ND |
| Isoleucine | g | 3.37 | 0.95 | ND |
| Leucine | g | 6.22 | 1.70 | ND |
| Lysine | g | 5.08 | 1.58 | ND |
| Methionine | g | 1.46 | 0.83 | ND |
| Methionine-cystine | g | 2.43 | 1.63 | ND |
| Phenylalanine | g | 3.94 | 1.13 | ND |
| Phenylalanine-tyrosine | g | 6.65 | 1.85 | ND |
| Threonine | g | 3.23 | 1.20 | ND |
| Tryptophan | g | 0.91 | 0.40 | ND |
| Valine | g | 4.00 | 1.23 | ND |
| L-Taurine | g | 0.60 | ND | ND |
| L-Carnitine | mg | 28.83 | ND | ND |
| *Crude fat* | g | 33.28 | 13.8 | ND |
| Linoleic acid | g | 7.76 | 2.80 | ND |
| Alpha-linoleic acid | g | 2.74 | ND | ND |
| ω-6/ω-3 FA ratio | | 3:1 | ND | 30:1 |
| *Minerals* | | | | |
| Calcium | g | 2.62 | 1.25 | 6.25 |
| Phosphorous | g | 2.33 | 1.00 | 4.00 |
| Ca:P ratio | | 1:1 | 1:1 | 2:1 |
| Potassium | g | 2.94 | 1.5 | ND |
| Sodium | g | 1.06 | 0.20 | ND |
| Chloride | g | 2.40 | 0.30 | ND |
| Magnesium | g | 0.46 | 0.15 | ND |
| Iron | mg | 94.48 | 10.00 | ND |
| Copper | mg | 5.99 | 1.83 | ND |
| Manganese | mg | 13.42 | 1.25 | ND |
| Zinc | mg | 46.24 | 20 | 286 |
| Iodine | mg | 0.37 | 0.25 | 2.75 |
| Selenium | mg | 0.16 | 0.08 | 0.5 |
| *Vitamins and others* | | | | |
| Vitamin A | IU | 2,697.37 | 1,250 | 62,500 |
| Vitamin D | IU | 201.80 | 125 | 750 |
| Vitamin E | IU | 88.89 | 12.5 | ND |
| Thiamine, Vit.B1 | mg | 5.48 | 0.56 | ND |
| Riboflavin, Vit.B2 | mg | 3.54 | 1.3 | ND |
| Niacin, Vit.B3 | mg | 17.78 | 3.4 | ND |
| Pantothenic acid, Vit.B5 | mg | 7.76 | 3.0 | ND |
| Pyridoxine, Vit.B6 | mg | 1.40 | 0.38 | ND |
| Folic acid, Vit.B9 | mg | 0.26 | 0.054 | ND |
| Cobalamin, Vit.B12 | mg | 0.01 | 0.007 | ND |

*(Continued)*

**Table 1.** (Continued)

| Nutrient | Units per 1000 kcal ME | Study Diet (Eurofins Analysis) | AAFCO K9 Adult Maintenance | |
| --- | --- | --- | --- | --- |
| | | | Minimum | Maximum |
| Choline | mg | 667.92 | 340 | ND |

Ingredients of Study Diet (https://v-dog.com/pages/vegan-dog-nutrition): Dried Peas, Pea Protein, Brown Rice, Oatmeal, Potato Protein, Sorghum, Canola Oil (preserved with Mixed Tocopherols), Natural Flavor, Suncured Alfalfa Meal, Brewers Dried Yeast, Dicalcium Phosphate, Flaxseeds, Millet, Calcium Carbonate, Lentils, Peanut Hearts, Quinoa, Sunflower Chips, Salt, Potassium Chloride, Choline Chloride, Taurine, Vitamins (Vitamin E Supplement, Vitamin A Supplement, Niacin Supplement, d-Calcium Pantothenate, Riboflavin Supplement, Vitamin D2 Supplement, Thiamine Mononitrate, Vitamin B12 Supplement, Pyridoxine Hydrochloride, Biotin, Folic Acid), Dried Carrots, Minerals (Ferrous Sulfate, Zinc Sulfate, Copper Sulfate, Sodium Selenite, Manganese Sulfate, Calcium Iodate), DL-Methionine, Dried Parsley, L-Ascorbyl-2-Polyphosphate (source of Vitamin C), preserved with Citric Acid, preserved with Mixed Tocopherols, Dried Celery, Dried Blueberries, Dried Cranberries, Dried Beets, Yucca Schidigera Extract, Dried Lettuce, L-Carnitine, Dried Watercress, Dried Spinach, Rosemary Extract

$Q_3$) provided by the reference laboratory (UC Davis Amino Acid Laboratory) [21] at baseline, intermediate, and endpoint (**S3 Table**), except tryptophan levels (which measured within the normal minimum to maximum reference interval derived through extrapolation).

Tryptophan levels measured below the first quartile provided by the UCD reference laboratory in 87% of dogs (13 of 15) at baseline while consuming a meat-based diet, 80% of dogs (12 of 15) at 6 months, and 40% of dogs (6 of 15) at 12 months while consuming K9PBN ($p = 0.01$). Since the reference intervals were provided as quartiles [21], and because tryptophan is an essential AA, we calculated the minimum to maximum reference interval for this AA using the quartiles provided by UC Davis, knowing the interquartile range (IQR) can be derived using standard conventions (S3 Table) [28]. The tryptophan levels at all time points

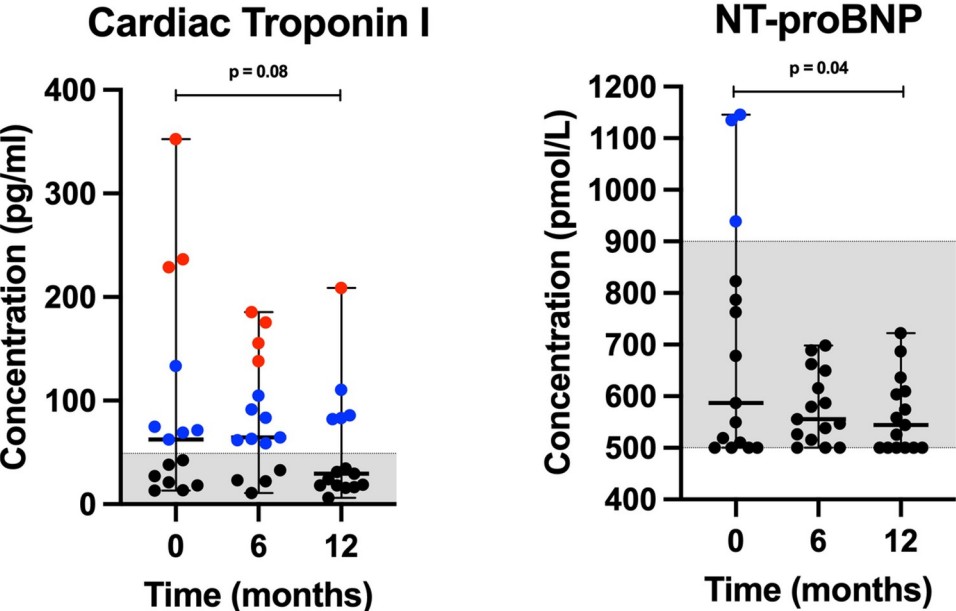

**Fig 2. Scatter plots of serum canine cardiac troponin I (cTnI), and serum N-terminal pro brain natriuretic peptide (NT-proBNP) concentrations in dogs at 0, 6 and 12 months.** The Vcheck measurement range for the NT-proBNP assay is 500–10,000 pmol/L, and the limit of detection (500 pmol/L) is reflected in 29% (13 of 45) of all the samples. Horizontal bars represent median values, while minimum and maximum values are shown by short horizontal lines. Black, blue, and red data points represent normal, equivocal, and abnormal values, respectively. The grey-shaded areas reflect normal levels for cTnI (0–49 pg/ml), and NT-proBNP (< 900 pmol/L).

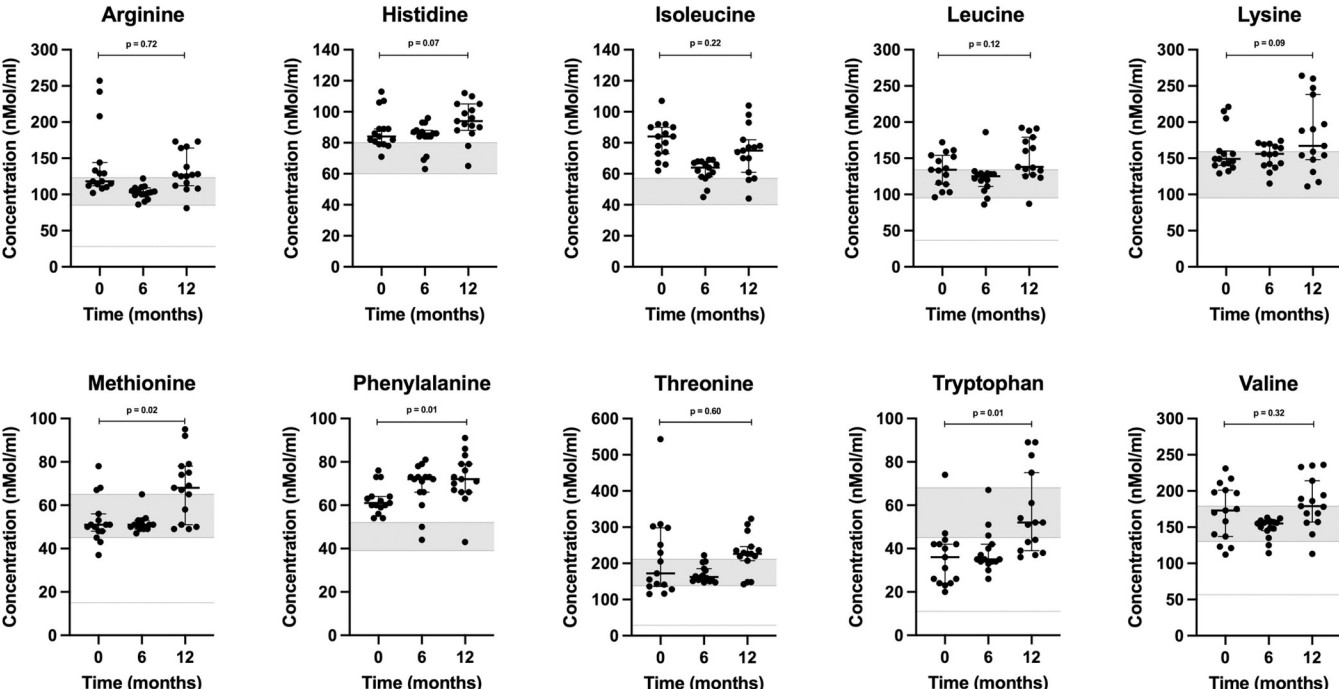

**Fig 3. Scatter plots of essential amino acids (arginine, histidine, isoleucine, leucine, lysine, methionine, phenylalanine, threonine, tryptophan, valine) concentrations in dogs at 0, 6 and 12 months.** Horizontal bars represent medians, while first and third quartiles are shown by short horizontal lines. The grey-shaded areas reflect the interquartile reference intervals ($Q_1$ –$Q_3$) provided by UCD. Lower dashed lines (arginine, leucine, methionine, threonine, tryptophan, and valine) reflect calculated minimum values derived by extrapolation (S3 Table).

were within the calculated (min.–max.) reference interval (11–103 nmol/ml). We found no statistically significant difference in neither L-taurine nor L-carnitine levels between baseline and endpoint, but their median values increased between 0 and 12 months (**Fig 4**).

## Serum vitamin concentrations

Serum vitamin A concentrations exhibited a relative increase (p = 0.01) during the trial, while median values stayed within the reference interval (**S4 Table**). Vitamin D (25-hydroxyvitamin D) levels were insufficient in 47% (7 of 15) at baseline, 7% (1 of 15), and 0% of the dogs at endpoint (**Fig 5**) with a statistically significant change over time (p = 0.004). Vitamin E levels measured above the reference interval at all three time points with no statistically significant change over time (p = 0.33).

Folate (p = 0.04) and cobalamin (p = 0.25) increased over time (**Fig 6**), when comparing serum values across the time points. Folate levels measured below the serum reference interval provided by the TAMU GI Laboratory in 40% (6 of 15), 7% (1 of 15), and 20% (3 of 15) of the dogs at 0, 6, and 12 months, respectively. We found no statistically significant differences in baseline versus endpoint values for neither folate (p = 0.09) nor cobalamin (p = 0.17).

## Discussion

In this study, we confirm that clinically healthy adult dogs maintain health when fed a nutritionally complete, commercially available, plant-based diet with pea protein as a main ingredient over a twelve-month period.

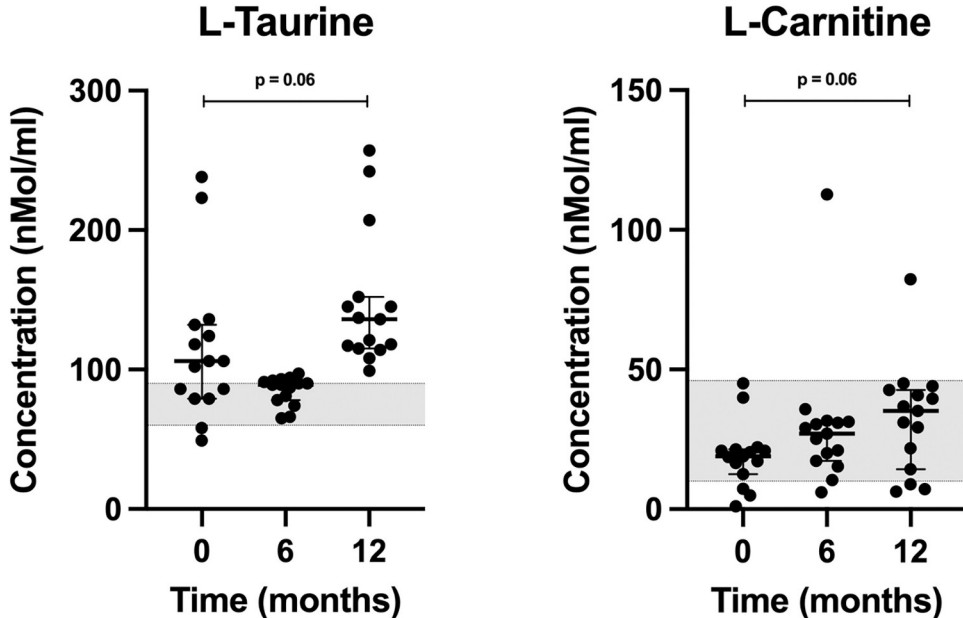

**Fig 4. Scatter plots of L-taurine, and L-carnitine concentrations in dogs at 0, 6 and 12 months.** Horizontal bars represent medians, while first and third quartiles are shown by short horizontal lines. The grey-shaded areas reflect reference intervals for L- taurine (60–90 nmol/ml) provided by the UCD Amino Acid Laboratory, and L-carnitine (10–46 uMol/L), as reported by Neumann, et al. (2007).

Our results align with previously published, short term, studies reporting on the outcomes in dogs fed K9PBN [4, 6, 14]. A recent study by Cavanaugh, *et al*, evaluated the same diet as used in our study (including peas as a primary protein source) in dogs over a shorter time period [14]. The authors did not detect any essential AA (or taurine) deficiencies, nor any clinically relevant changes in hematological, serum biochemical or echocardiographic parameters

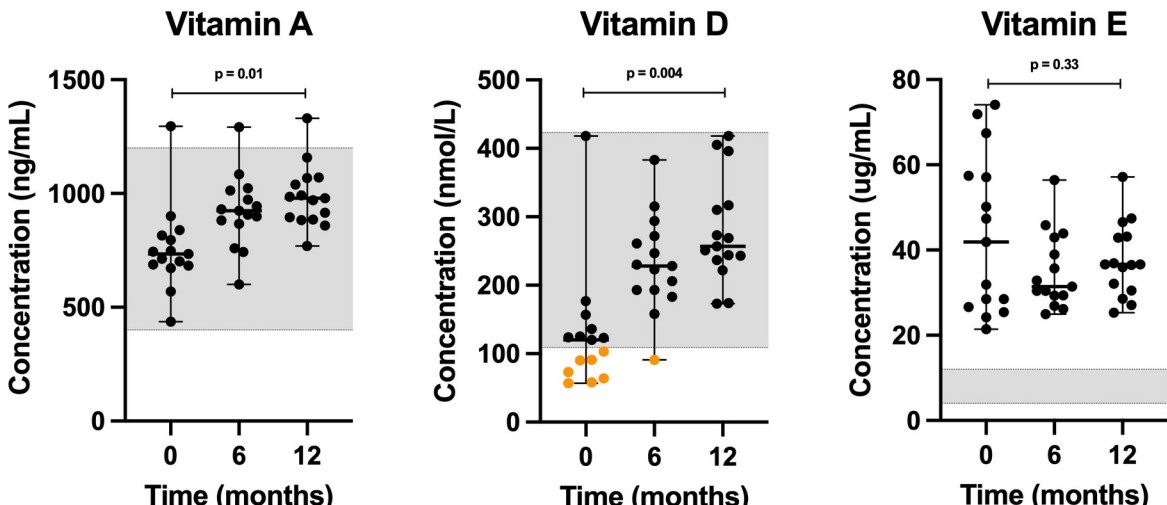

**Fig 5. Scatter plots of serum lipid-soluble vitamins, including vitamin A, 25-hydroxyvitamin D, and vitamin E (alpha tocopherol) concentrations in dogs at 0, 6 and 12 months.** Horizontal bars represent median values, while minimum and maximum values are shown by short horizontal lines. Values that are either within the MSU Veterinary Diagnostic Laboratory's reference interval (grey-shaded area), or above, are depicted by black dots, while concentrations that are below the reference interval are depicted by orange dots.

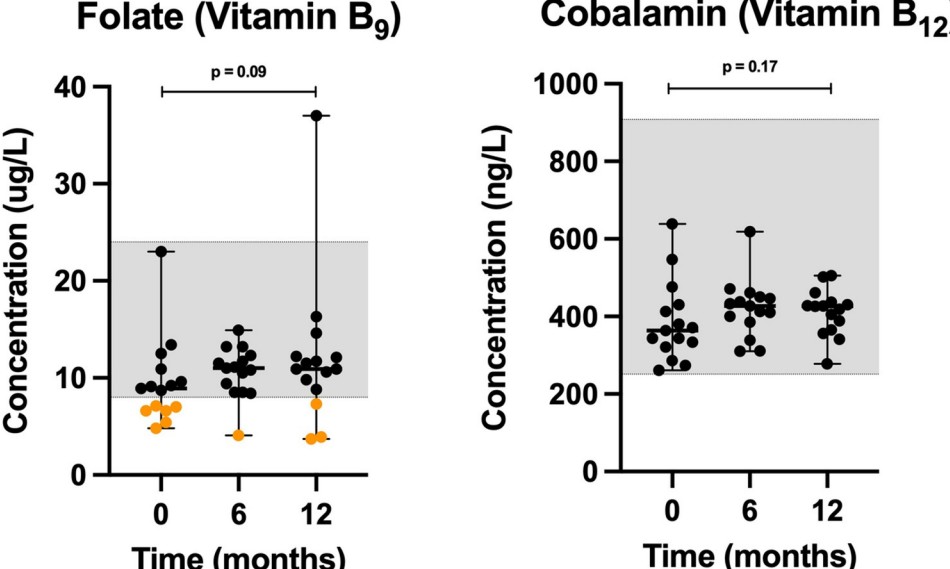

**Fig 6. Scatter plots of serum water-soluble vitamins, including vitamin B$_9$ and vitamin B$_{12}$ concentrations in dogs at 0, 6 and 12 months.** Horizontal bars represent median values, while minimum and maximum values are shown by short horizontal lines. Values that are either within the TAMU GI Laboratory's reference interval (grey-shaded area), or above, are depicted by black dots, while concentrations below the reference interval are depicted by orange dots.

after feeding K9PBN to dogs. A 2009 study on canine endurance athletes, which evaluated a plant-based diet in sled dogs over 16 weeks, further supports our findings. This study found that sled dogs maintained hematological parameters and high performance when fed a custom-made meat-free diet [6]. K9PBN is not a new concept when viewed in a global context where many dogs maintain health on vegetarian diets with minimal to no animal-derived ingredients. Still, the scientific study of health outcomes in response to K9PBN has been limited.

Nutrient analysis of the dry food used in this study confirmed compliance with AAFCO nutrient profiles for adult canine maintenance [16]. Additionally, the level of glyphosate (0.18 mg/kg), a widely used herbicide, was measured in the kibble, and found to be significantly below the ADI for humans in the US (1.75 mg/kg) and the EU (0.5 mg/kg) [18]. We also documented that the body weight of dogs that switched to K9PBN remained stable, while body condition scores trended downwards in overweight/obese dogs (Fig 1). Plant-based nutrition is known to reduce body fat in people through various mechanisms, including the lower caloric density of many whole plant foods, impact on energy balance through gut microbiome modulation, increased post-prandial metabolism (diet-induced thermogenesis), etc. [29, 30].

Consumer concerns about heart health and potential risk of diet-associated DCM (da-DCM) when feeding non-traditional dog foods warrants careful consideration. We evaluated cardiac biomarkers retrospectively in all dogs at 0, 6, and 12 months. We did not observe a statistically significant difference in neither cTnI nor NT-proBNP levels across the three time points, but we did observe a downward trend in both biomarkers (Fig 2). All dogs were clinically healthy; however, the cTnI levels were increased in three dogs at baseline, compared to only one dog at endpoint. We recognize these biomarkers are not specific for DCM, however, these findings may alleviate some concerns about the safety of K9PBN. Plant-based nutrition is a pillar of lifestyle medicine and of central importance to human heart health, specifically prevention and reversal of atherosclerosis [31, 32].

The essential AAs arginine, histidine, isoleucine, leucine, lysine, methionine, phenylalanine, threonine, and valine measured within or above normal reference intervals using the first and third quartiles (50% of expected data points) at all time points with an upward trend (Fig 3). Tryptophan levels measured within the reference interval of minimum to maximum values (100% of expected data points). These findings confirm the K9PBN study diet provided all essential AAs exogenously. Additionally, we measured L-taurine (plasma) and L-carnitine (serum) levels and observed a statistically non-significant increase in these nutrients when comparing baseline to endpoint values (Fig 4). We acknowledge the inherent discrepancy between serum and tissue L-carnitine levels. However, these were clinically healthy companion dogs and collection of myocardial biopsies would not be a relevant consideration in this context. We did not measure taurine levels in whole blood or urine, which is a study limitation. The third-party laboratory did not provide reference values for canine L-carnitine serum levels, however, dogs served as their own controls in this study and our findings are consistent with L-carnitine (plasma) levels reported in other clinically healthy dogs with a mean value of 24 uMol/L (range 10–46 uMol/L) [33]. Taurine and L-carnitine are important to myocardial health, and both have been used as nutritional supplements in conjunction with pharmaceutical intervention to address canine DCM [22]. Taurine is derived from sulfur-containing AAs (methionine and cysteine) and considered a conditionally non-essential AA in dogs. As such, taurine deficiency may render some dog breeds more vulnerable to nutritional dilated cardiomyopathy (nDCM) [34, 35]. Taurine impacts the regulation of tissue calcium concentrations and free radical inactivation [22]. L-carnitine is a derivative from essential AAs lysine and methionine and important for long-chain fatty acid oxidation, which is the main energy source for cardiomyocytes in heart health [36]. The mitochondrial fatty acid oxidation capacity is reduced in heart failure where the myocardium increasingly relies on glucose and ketone bodies as alternative sources of energy [36]. Carnitine deficiency has been associated with DCM in humans and dogs, however, elevation in circulating carnitine has also been reported in DCM patients, which may be a compensatory response in the failing myocardium to increase fatty acid oxidation [36]. In humans with heart failure from DCM, L-carnitine supplementation may improve survival time [37]. Carnitine supplementation has also been observed to support canine heart health [38].

Almost half (7 of 15) of the dogs presented with insufficient levels of 25-hydroxyvitamin D at baseline (Fig 5). Vitamin D insufficiency in clinically healthy dogs has been reported. One US study measured vitamin D serum levels in a sample of 320 clinically healthy dogs from breed clubs across the US consuming meat-based diets, and found that 85% of dogs were vitamin D insufficient (while 1% of the dogs was vitamin D deficient) [39]. These authors reported that vitamin D levels remained insufficient in approximately 75% of dogs irrespective of supplementation (fish oils). In contrast, we found that vitamin D levels normalized in most dogs at six months (6 of 7 vitamin D insufficient dogs), and all dogs at 12 months, which was achieved with K9PBN alone and without supplement use (Fig 5). Interestingly, a recent communication highlighted that the results from Sharp, *et al.* (2015), as compared to data reported by the MSU VDL, may be explained by differences in analytical methods between the laboratories [40].

The MSU VDL reported that an adequate serum vitamin E concentration for adult dogs is between 4 and 12 ug/mL (Fig 5). The vitamin E levels in this study were more than adequate, and the relative increase in concentrations is considered unremarkable without cause for concern.

Cobalamin (vitamin $B_{12}$) and folate (vitamin $B_9$) were measured at 0, 6, and 12 months. While cobalamin measured within the normal reference interval at all three time points, folate levels measured below the serum reference interval provided by the laboratory (TAMU GI

Lab, College Station, TX) in 6, 1, and 3 of the dogs at 0, 6, and 12 months, respectively (Fig 6). None of these dogs presented with clinical signs that would indicate folate insufficiency, and dogs served as their own controls. Here, the number of folate insufficient dogs decreased by 50% at endpoint after switching from a meat-based diet to K9PBN without supplement use. Supplementation with folate in these cases would have been empirical since the clinical benefits have not been established in dogs without clinical signs, and the sensitivity and specificity of this diagnostic assay has not been established [41]. A team of board-certified nutritionists have reported a different normal reference interval for plasma folate concentrations with a lower minimum value (4–26 ng/mL) [42], suggesting our measurements at all time points could be within normal range. Comparatively, the normal reference interval for folate levels in humans is 6–20 ng/mL, while 3–5.9 ng/mL reflects possible deficiency, and < 3 ng/mL deficiency [43].

We recognize that data from a clinical feeding trial involving client-owned dogs is reliant upon client compliance with instructions to feed only the prescribed study diet/treats. We conducted informal monthly phone interviews with clients to monitor the well-being of canine participants between the quarterly on-campus appointments. During the phone interviews, it was moreover confirmed that all canine participants were fed only the prescribed plant-based study diet/treats (i.e., no foods or treats containing animal-based products). We acknowledge that the phone interviews were not conducted using a validated formal questionnaire.

The study was also limited by structural barriers that meant echocardiographic exams were not obtained. Still, we evaluated cardiac biomarkers, which reflected this nutritional intervention did not impact cardiac status adversely (based on blood biomarkers only) and may have had a positive impact on overall heart health in some dogs. This finding contrasts concerns about using pea protein as a main ingredient [13]. We recognize that a comprehensive cardiac assessment would include echocardiography, and that the absence of such data is a study limitation. We plan to explore these novel findings in combination with echocardiography in future studies on K9PBN.

Significant strengths of this study include its duration and comprehensiveness. Also, all blood nutritional markers and cTnI analyses were performed as fee-for-service by accredited third-party laboratories, which is an additional strength of this study. To our knowledge, this is the longest clinical feeding trial evaluating a nutritionally complete plant-based diet in companion dogs. Furthermore, we measured a significantly greater number of clinical, nutritional, and hematological parameters than any other K9PBN study. Our study design goes far beyond AAFCO feeding trial guidelines [16], which recommend a minimum study period of six months and eight dogs with the health exams performed by a licensed veterinarian and assessment of four blood parameters, including hematocrit, hemoglobin, albumin, and alkaline phosphatase.

The ability to ensure individuals maintain health on non-animal-based food sources is a key component of the ongoing societal shift towards sustainable and more equitable food systems in a global health context. A recent publication in the American Journal of Cardiology also points out that efforts to promote nutrition equity through the emphasis on plant-based nutrition is a moral imperative of the medical profession [44]. With a population of nearly 90 million dogs, the US has the most canine companions per capita worldwide [45]. Dogs have a wealth of positive implications for human health; however, canine populations also translate into a substantial amount of consumption and waste production. A 2017 study estimated dog and cat food production is responsible for up to 30% of the total environmental costs from industrial food animal production [46]. This needs to also be viewed in light of the total cumulative adverse impacts of industrial food animal production and the tremendous potential that could be derived from phasing out this industry with respect to reduction of total global greenhouse gas emissions [47].

Comparatively, humans require nine essential AAs from exogenous sources (histidine, iso-leucine, leucine, lysine, methionine, phenylalanine, threonine, tryptophan, and valine.) Dogs have an additional requirement for arginine, while cats have yet another requirement for tau-rine [48]. The common view that plant proteins are incomplete and inferior to animal proteins is a fallacy from a nutrition science perspective since protein from non-animal sources provide sufficient essential AAs to maintain health. It is well-established that plant proteins are ade-quate for meeting essential AA requirements for humans since the body possesses an inherent capacity to mix and match AAs to meet required proportions [49]. One study evaluated mac-ronutrient and AA compositions in high-quality and novel protein sources, hereunder pea, potato, faba bean, soy, and dried yeast in reference to canine nutrition [50]. The protein quality and high essential AA digestibility of these ingredients make them viable protein sources for use in dog food alongside ingredients with high sulfur-containing AAs, such as cereal grains to provide sufficient methionine [50].

Studies by Knight, *et al.*, have assessed different aspects of feeding vegan and vegetarian diets to companion animals [4, 7]. A 2022 study aimed to evaluate the nutritional soundness and quality of pet foods globally. The study reported on superior standards for plant-based diets at nearly all stages examined, throughout the design, manufacturing, transportation and storage phases [7]. Two studies reported on quality control concerns in reference to some commercial vegetarian diets [51, 52]. A recent review highlighted the need for collaboration between stakeholders within the pet food industry, veterinary medical community, and con-sumers to advance companion animal care sustainably [53].

Canine nutrition using novel protein sources (including peas) has been a subject of recent interest in light of da-DCM cases reported in atypical dog breeds in the US [54]. It is important to assess potential risks of nutrient deficiencies (e.g., assessment of protein (essential AAs), vitamin $B_{12}$, vitamin $D_3$, taurine, L-carnitine, etc.) when using novel protein sources. Notably, the relative uptick in da-DCM cases in the US has not been reported at a comparable scale in Europe or elsewhere, and focus is shifting towards assessment of potential culprits related to ingredient contaminants versus specific nutrient deficiencies. The veterinary community is in need of evidence-based K9PBN research at the intersection between clinical cardiology and nutrition to define clinical health outcomes in dogs transitioned to a diet produced indepen-dent of industrial food animal production. This is important because existing food systems are undergoing change as a result of increasing costs of animal-derived ingredients combined with the escalating climate crisis, thus supporting studies on more cost-effective and sustain-able options for feeding companion dogs. The present study provides an important stepping stone in that direction.

## Supporting information

**S1 Table. Complete blood count and blood chemistry analyses in dogs consuming meat-based diets (baseline) versus plant-based nutrition (6 and 12 months).** Values reflect median (minimum—maximum).
(DOCX)

**S2 Table. Urinalysis in dogs consuming meat-based diets (baseline) versus plant-based nutrition (6 and 12 months).** Values refer to median (minimum—maximum). N/A: not applicable (since each row has zero difference, which precludes calculation of a paired test). Semi-quantitative UA parameters were tabulated as '0' (normal/negative/not detected or <1/HPF), '1' (trace), or the highest reported value (e.g., 1-5/HPF and 6-20/HPF were tabulated as '5' and '20', respectively), where HPF is high power field. Urine samples were collected via

cystocentesis where iatrogenic microscopic hematuria is to be expected.
(DOCX)

**S3 Table. Nutrient analysis including plasma amino acid and serum L-carnitine concentrations in dogs consuming meat-based diets (baseline) versus plant-based nutrition (6 and 12 months).** Values refer to median (first quartile—third quartile). Reference values for first and third quartiles were provided by UCD. We requested reference intervals (minimum to maximum) from this laboratory, but were informed they were not available.
(DOCX)

**S4 Table. Nutrient analysis of serum vitamin concentrations in dogs consuming meat-based diets (baseline) versus plant-based nutrition (6 and 12 months).** Values refer to median (minimum—maximum).
(DOCX)

## Acknowledgments

We thank the organizing team behind the Plant-Based Dog Food Health Study Initiative in Los Angeles, California for facilitating the fundraising efforts that made this study a reality. We acknowledge support from the WesternU College of Veterinary Medicine Office for Research, the True One Medicine Initiative, and colleagues at the WesternU PHC who provided technical support. We also acknowledge BioNote, Inc. for providing the Vcheck200 Analyzer and the canine NT-proBNP assays. Our team is grateful for the participating dogs and clients. The authors declare no financial conflict of interest. The study was completed independent of sponsorship from V-Dog. The Kind Kibble and Wiggle Biscuits were purchased from V-Dog using research funding.

## Author Contributions

**Conceptualization:** Annika Linde, Tonatiuh Melgarejo.

**Data curation:** Annika Linde, Maureen Lahiff, Theros T. Ng, Tonatiuh Melgarejo.

**Formal analysis:** Annika Linde, Maureen Lahiff, Theros T. Ng, Tonatiuh Melgarejo.

**Funding acquisition:** Annika Linde, Tonatiuh Melgarejo.

**Investigation:** Annika Linde, Adam Krantz, Nathan Sharp, Theros T. Ng, Tonatiuh Melgarejo.

**Methodology:** Annika Linde, Maureen Lahiff, Theros T. Ng, Tonatiuh Melgarejo.

**Project administration:** Annika Linde.

**Resources:** Annika Linde, Tonatiuh Melgarejo.

**Software:** Theros T. Ng.

**Supervision:** Annika Linde, Tonatiuh Melgarejo.

**Validation:** Annika Linde, Tonatiuh Melgarejo.

**Visualization:** Annika Linde, Theros T. Ng.

**Writing – original draft:** Annika Linde.

**Writing – review & editing:** Annika Linde, Maureen Lahiff, Adam Krantz, Nathan Sharp, Theros T. Ng, Tonatiuh Melgarejo.

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
