## [Decision Letter · Decision Letter 0]

7 Aug 2023

PONE-D-23-18094Domestic dogs maintain clinical, nutritional, and hematological health outcomes when fed a commercial plant-based diet for a yearPLOS ONE

Dear Dr. Melgarejo,

Thank you for submitting your manuscript to PLOS ONE. After careful consideration, we feel that it has merit but does not fully meet PLOS ONE’s publication criteria as it currently stands. Therefore, we invite you to submit a revised version of the manuscript that addresses the points raised during the review process.

 Please submit your revised manuscript by Sep 09 2023 11:59PM. If you will need more time than this to complete your revisions, please reply to this message or contact the journal office at plosone@plos.org. Please include the following items when submitting your revised manuscript:A rebuttal letter that responds to each point raised by the academic editor and reviewer(s). You should upload this letter as a separate file labeled 'Response to Reviewers'.A marked-up copy of your manuscript that highlights changes made to the original version. You should upload this as a separate file labeled 'Revised Manuscript with Track Changes'.An unmarked version of your revised paper without tracked changes. You should upload this as a separate file labeled 'Manuscript'.

We look forward to receiving your revised manuscript.

Kind regards,

Juan J Loor

Academic Editor

PLOS ONE

Journal Requirements:

AL & TM received research funding from Plant-Based Dog Food Health Study Initiative in Los Angeles, California, WesternU College of Veterinary Medicine Office for Research, the True One Medicine Initiative at WU.

3. Thank you for providing the following Funding Statement:  

AL & TM received research funding from Plant-Based Dog Food Health Study Initiative in Los Angeles, California, WesternU College of Veterinary Medicine Office for Research, the True One Medicine Initiative at WU.

We note that one or more of the authors is affiliated with the funding organization, indicating the funder may have had some role in the design, data collection, analysis or preparation of your manuscript for publication; in other words, the funder played an indirect role through the participation of the co-authors. 

If the funding organization did not play a role in the study design, data collection and analysis, decision to publish, or preparation of the manuscript and only provided financial support in the form of authors' salaries and/or research materials, please review your statements relating to the author contributions, and ensure you have specifically and accurately indicated the role(s) that these authors had in your study in the Author Contributions section of the online submission form. Please make any necessary amendments directly within this section of the online submission form.  Please also update your Funding Statement to include the following statement: “The funder provided support in the form of salaries for authors [insert relevant initials], but did not have any additional role in the study design, data collection and analysis, decision to publish, or preparation of the manuscript. The specific roles of these authors are articulated in the ‘author contributions’ section.” 

If the funding organization did have an additional role, please state and explain that role within your Funding Statement. 

Please also provide an updated Competing Interests Statement declaring this commercial affiliation along with any other relevant declarations relating to employment, consultancy, patents, products in development, or marketed products, etc.  

4. We noted in your submission details that a portion of your manuscript may have been presented or published elsewhere. Please clarify whether this publication was peer-reviewed and formally published. If this work was previously peer-reviewed and published, in the cover letter please provide the reason that this work does not constitute dual publication and should be included in the current manuscript.

Reviewers' comments:

Reviewer's Responses to Questions

**Comments to the Author**

1. Is the manuscript technically sound, and do the data support the conclusions?

Reviewer #1: Partly

Reviewer #2: Yes

Reviewer #3: Yes

2. Has the statistical analysis been performed appropriately and rigorously? 

Reviewer #1: I Don't Know

Reviewer #2: Yes

Reviewer #3: Yes

3. Have the authors made all data underlying the findings in their manuscript fully available?

Reviewer #1: Yes

Reviewer #2: Yes

Reviewer #3: Yes

4. Is the manuscript presented in an intelligible fashion and written in standard English?

Reviewer #1: Yes

Reviewer #2: Yes

Reviewer #3: Yes

5. Review Comments to the Author

Reviewer #1: This project aimed to look at the health of dogs fed a K9PBD for 12 months by evaluating overt clinical health, blood work, and nutrient biomarkers. The investigators noted no statistical change over time within the sample population following long term ingestion of a non-animal protein based diet. Secondarily, the authors were notably trying to highlight the lack of change in cardiac biomarkers when fed a pea protein diet to reduce the fears of dietary-induced cardiomyopathy.

The reviewer believes that this study does not meet rigorous research standards due to many omissions, including nutrition information each individual subjects' diets before the study and the feeding recommendations during the study. In addition, sweeping comments regarding cardiac health are questionable as true evaluation of cardiac function was not performed; cardiac biomarkers have significant limitations and are not recommended for screening in a general healthy dog population. This study was simple and should be stated as such- maybe a brief communication rather then a publication. The reviewer would not have read this paper after reading the MM; the author should request critical appraisal prior to submission.

Introduction: The reviewer notes that in this introduction, the concern over cardiomyopathy in dogs eating non-traditional foods was not addressed as this was in large part an important portion of the study design and discussion. If the audience for this publication is veterinary health professionals, whom are making the recommendations to pet owners regarding diet, it is essential to highlight the "whys" of this project.

Line 80- Conclusions should not be in the introduction

Study Design: This is a small study population (n=17) and a biased population as people enrolled were all participants in health sciences; if many of these dogs were owned by the veterinary population, stronger bias exists. As the veterinarian evaluating the dogs was not blinded, this should be duly noted. What were the inclusion/exclusion criteria? The client questionnaire should be in supplementary material and noted that it is not validated. No mention of the feeding recommendations is noted- this is essential as the author addresses BCS and body weight. This is an oversight that cannot go overlooked.

Discussion: The reviewer notes areas in the discussion which are superfluous and areas that are considerably lacking. Lines 347-353, which in fact states that dogs can maintain health over time on a meat free diet (which was published in 2009). The current study, therefore, does not provide additional information to the reader. Lines 358-359 the author highlights the reduction in BCS- as no feeding recommendations were noted nor compared with previous kcal, protein, fat, carbohydrate consumption this finding is questionable to be a result of a plant based diet. The change simply may have been a reduction in calories ingested at the bare minimum. The diets ingested prior to the study were not mentioned- therefore the reader has no idea how to compare this change. No mention of muscle condition scoring is noted; did the dogs loose/maintain muscle mass? Were their changes in exercise/environment/etc? Since the owners were not blinded, were they also biased in their reporting? Evaluation of biomarkers is very questionable as they have limited clinical screening use in healthy dogs. The reviewer does not agree with the conclusion that the study findings alleviate any of the cardiomyopathy concerns as no true screening (VHS, Echo) was completed prior to diet change or after 1 year on the diet. To interpret the biomarkers, the prevalence of disease in the population needs to be considered as well as their considerable limitations in a dog with occult DCM. The reviewer is struggling to understand the discussion on Vitamin D- Lines 414-416 discuss the differences in analytical methods as possibly being the reason why the greater population of dogs were notably D deficient. If this is the case, then the comparison to the large population is not applicable. The reviewer is just questioning the comments in this section as these comments appear to negate the findings. Line 446 just simply mentions the pea protein concerns- and the author is trying to assuage their audience of its use in dogs, why is there so little information in the discussion. To the reviewer, it appears as an afterthought and not worthy of my time as a reader. 450-458 simply states what the author has already said- unacceptable to be restated. 460-471 is simply just more words to fill the page and can be stated in a simple sentence in the introduction rather than a paragraph in the discussion. Following a brief discussion of findings, the reader is again presented with a paragraph on PBD- the author again needs to keep these thoughts together rather than spread them out through the discussion in a haphazard fashion.

Reviewer #2: The authors evaluated the effect of a pea-based protein diet in dogs, detecting its safety through clinical information and various laboratory parameters. The authors themselves highlighted and made it clear that the main limitation of the study is its duration of only one year. In this regard, it is difficult to extrapolate that these results are conclusive when considering longer periods, and as the authors mention, further studies will be necessary for confirmation. Another significant limitation mentioned includes cardiac evaluation, as the markers used may not adequately reflect cardiac function, and the time for the development of nutritional cardiopathy is relatively short. As these limitations have been adequately discussed, I believe the manuscript should be published, as it truly opens perspectives for dietary changes in dogs, providing evidence to support this change. Some minor points are included in the sequence.

Introduction

Lines 48-52: The authors used two initial references that provide broad information about immunity and nutrition in reference to dogs, but current knowledge does not allow for such extrapolations.

Line 78: The authors should bring the meaning of the acronym AAFCO in full.

MATERIALS AND METHODS

Study design

The main concern of this reviewer is whether, in fact, the dogs received only plant-based food, that is, whether the tutors did not provide any other type of food during the study.

Reviewer #3: The manuscript is very well and clearly written. I have only two comments:

- Would it be possible to analyse urine for taurine? Urinary taurine is considered by many researchers to be the most reliable measure of taurine status and would add to the understanding of whether the dogs are retaining or at risk of becoming depleted in taurine

- The Discussion is quite long, largely because the results are being repeated in the text. I recommend that the authors make efforts to reframe the Discussion.

6. PLOS authors have the option to publish the peer review history of their article (what does this mean?). If published, this will include your full peer review and any attached files.

Reviewer #1: No

Reviewer #2: No

Reviewer #3: **Yes: **Anne Marie Bakke, Prof. Dr.med.vet.

---

## [Author Response · Author response to Decision Letter 0]

5 Sep 2023

Reviewer #1: This project aimed to look at the health of dogs fed a K9PBD for 12 months by evaluating overt clinical health, blood work, and nutrient biomarkers. The investigators noted no statistical change over time within the sample population following long term ingestion of a non-animal protein based diet. Secondarily, the authors were notably trying to highlight the lack of change in cardiac biomarkers when fed a pea protein diet to reduce the fears of dietary-induced cardiomyopathy.

The reviewer believes that this study does not meet rigorous research standards due to many omissions, including nutrition information each individual subjects' diets before the study and the feeding recommendations during the study. In addition, sweeping comments regarding cardiac health are questionable as true evaluation of cardiac function was not performed; cardiac biomarkers have significant limitations and are not recommended for screening in a general healthy dog population. This study was simple and should be stated as such- maybe a brief communication rather then a publication. The reviewer would not have read this paper after reading the MM; the author should request critical appraisal prior to submission.

Thank you for these comments. The authors maintain that the significant extent of data justifies a full-length publication and would not seem suitable for a brief communication. It is correct that the specific nutrition information was not included for individual dogs, because that was not within the scope of the study. Our goal was simply to study dogs who transitioned from a meat-based diet to a plant-based diet. We were not looking to study the specific nutrition information in the meat-based diets at baseline. In our assessment, the study limitations have already been mentioned in the manuscript. It is correct that cardiac biomarker analysis in the absence of echocardiography is far from ideal, however, we had already stated that as a limitation in the original manuscript. 

Introduction: The reviewer notes that in this introduction, the concern over cardiomyopathy in dogs eating non-traditional foods was not addressed as this was in large part an important portion of the study design and discussion. If the audience for this publication is veterinary health professionals, whom are making the recommendations to pet owners regarding diet, it is essential to highlight the "whys" of this project.

Thank you for this feedback. It is correct that a complete cardiac exam, including echocardiography, would have been ideal in context of this study. Having said that, it is important to note that most feeding trials would not include an echocardiographic exam. In reality, the AAFCO guidelines for feeding trials have much fewer requirements (line 461-463: “recommend a minimum study period of six months and eight dogs with the health exams performed by a licensed veterinarian and assessment of four blood parameters, including hematocrit, hemoglobin, albumin, and alkaline phosphatase.”) than what we decided to have included for the purpose of this particular feeding trial.

Line 80- Conclusions should not be in the introduction

A summary statement is generally included at the end of an introduction. As such, we have modified the wording to say, ‘We showed that clinically healthy, adult, dogs can maintain health over a longer period of time when using a complete K9PBN approach.”

Study Design: This is a small study population (n=17) and a biased population as people enrolled were all participants in health sciences; if many of these dogs were owned by the veterinary population, stronger bias exists. As the veterinarian evaluating the dogs was not blinded, this should be duly noted. What were the inclusion/exclusion criteria? The client questionnaire should be in supplementary material and noted that it is not validated. No mention of the feeding recommendations is noted- this is essential as the author addresses BCS and body weight. This is an oversight that cannot go overlooked.

Thank you for these important comments. It is correct that dogs enrolled in the study were volunteered by WesternU student, staff, faculty, which is a biased population. We have already stated that the dogs were prospectively identified by emailing all students, staff, and faculty at Western University of Health Sciences (WesternU) in California (line 90-91). We have modified the text to state that the veterinarian evaluating the dogs was not blinded (line 99), and that we used a non-validated questionnaire (line 103). It has furthermore been added to the text (line 103-104), that participating dogs were fed by “following the manufacturer’s feeding recommendations”. 

Discussion: The reviewer notes areas in the discussion which are superfluous and areas that are considerably lacking. Lines 347-353, which in fact states that dogs can maintain health over time on a meat free diet (which was published in 2009). The current study, therefore, does not provide additional information to the reader. Lines 358-359 the author highlights the reduction in BCS- as no feeding recommendations were noted nor compared with previous kcal, protein, fat, carbohydrate consumption this finding is questionable to be a result of a plant based diet. The change simply may have been a reduction in calories ingested at the bare minimum. The diets ingested prior to the study were not mentioned- therefore the reader has no idea how to compare this change. No mention of muscle condition scoring is noted; did the dogs loose/maintain muscle mass? Were their changes in exercise/environment/etc? Since the owners were not blinded, were they also biased in their reporting? Evaluation of biomarkers is very questionable as they have limited clinical screening use in healthy dogs. The reviewer does not agree with the conclusion that the study findings alleviate any of the cardiomyopathy concerns as no true screening (VHS, Echo) was completed prior to diet change or after 1 year on the diet. To interpret the biomarkers, the prevalence of disease in the population needs to be considered as well as their considerable limitations in a dog with occult DCM. The reviewer is struggling to understand the discussion on Vitamin D- Lines 414-416 discuss the differences in analytical methods as possibly being the reason why the greater population of dogs were notably D deficient. If this is the case, then the comparison to the large population is not applicable. The reviewer is just questioning the comments in this section as these comments appear to negate the findings. Line 446 just simply mentions the pea protein concerns- and the author is trying to assuage their audience of its use in dogs, why is there so little information in the discussion. To the reviewer, it appears as an afterthought and not worthy of my time as a reader. 450-458 simply states what the author has already said- unacceptable to be restated. 460-471 is simply just more words to fill the page and can be stated in a simple sentence in the introduction rather than a paragraph in the discussion. Following a brief discussion of findings, the reader is again presented with a paragraph on PBD- the author again needs to keep these thoughts together rather than spread them out through the discussion in a haphazard fashion.

Thank you for providing this additional feedback. In reality, this study provides additional information to the reader since we conducted a twelve-month study and used many more parameters than the 2009 study. As mentioned, an in-depth nutrition evaluation of the meat-based diets used at baseline was not within the scope of this study. The clients fed the dogs according to the manufacturer’s recommendation, which has been added. It is correct that the change in BCS may simply reflect a difference in caloric intake. We are not making specific claims to state what is the specific reason, but simply including the relative change in BCS as an observation. It is correct that muscle condition scores were not recorded. In hindsight, this would have been valid to add. The dogs were not subject to any major environmental (or exercise) changes. The fact that clients were not blinded is not of major significance here since we did not use a cross-over design. Assessment of biomarkers involves certain inherent limitations. We have clearly acknowledged in the text what we perceive as the major study limitations. The lack of comprehensive cardiac evaluations is already listed as a study limitation. Still, we have included an additional sentence to highlight this limitation (line 448, 450-451). It should be noted that radiography (VHS) would not be a sensitive tool to screen for DCM. In reference to the concern about the discussion as it pertains to vitamin D, it is noted that MSU is considered the main reference laboratory for this nutritional parameter (vitamin D) in the US. So, we are simply including a comparison to another study that used a different reference laboratory in the UK. Therefore, this comparison doesn’t negate our findings. The mentioning of pea protein as a main ingredient is of interest since some papers have highlighted peas (and pulses) as ‘less desirable’ ingredients. Again, it is not within the scope of this manuscript to discuss pea protein in any greater detail. We find that it is key to highlight the relative length and comprehensiveness of the study. While the bigger One Health picture is a politically charged topic for some readers, we find that veterinary clinician-scientists have a moral responsibility to discuss such topics. Consequently, we find that this content (lines 460-471) serves a very important purpose.

Reviewer #2: The authors evaluated the effect of a pea-based protein diet in dogs, detecting its safety through clinical information and various laboratory parameters. The authors themselves highlighted and made it clear that the main limitation of the study is its duration of only one year. In this regard, it is difficult to extrapolate that these results are conclusive when considering longer periods, and as the authors mention, further studies will be necessary for confirmation. Another significant limitation mentioned includes cardiac evaluation, as the markers used may not adequately reflect cardiac function, and the time for the development of nutritional cardiopathy is relatively short. As these limitations have been adequately discussed, I believe the manuscript should be published, as it truly opens perspectives for dietary changes in dogs, providing evidence to support this change. Some minor points are included in the sequence.

We are grateful that Reviewer #2 recognizes that we have fully acknowledged the limitations of the study, and find that it is suitable for publication. Although this study is not a multi-year investigation, it is the longest study on this particular topic to date. 

Introduction

Lines 48-52: The authors used two initial references that provide broad information about immunity and nutrition in reference to dogs, but current knowledge does not allow for such extrapolations.

Yes, that this correct. We have deleted the word ‘canine’ from this sentence to address the concern that these references are not specific to canine health.

Line 78: The authors should bring the meaning of the acronym AAFCO in full.

Thank you for this comment. The sentence has been modified to include ‘Association of American Feed Control Officials’.

MATERIALS AND METHODS

Study design

The main concern of this reviewer is whether, in fact, the dogs received only plant-based food, that is, whether the tutors did not provide any other type of food during the study.

This is another important comment. Clinical research involving client-owned dogs requires a high degree of trust that clients follow specific feeding instructions. Unlike a feeding trial with laboratory beagles in a highly controlled environment, the canine study subjects in this project lived with the clients and they ultimately had full control of what these dogs would be consuming. Having said that, the participating dogs in this study were voluntarily enrolled by individuals in a higher education environment (including students, staff, faculty) and we therefore trust that they were following our instructions. 

Reviewer #3: The manuscript is very well and clearly written. I have only two comments:

We are thankful to Reviewer #3 for the positive feedback on the manuscript.

- Would it be possible to analyse urine for taurine? Urinary taurine is considered by many researchers to be the most reliable measure of taurine status and would add to the understanding of whether the dogs are retaining or at risk of becoming depleted in taurine

While it is possible to analyze urine for taurine, we only analyzed blood. Unfortunately, we do not have the ability to perform additional analyses at this point in time. In future feeding trials, we will take this important feedback into consideration. Many thanks!

- The Discussion is quite long, largely because the results are being repeated in the text. I recommend that the authors make efforts to reframe the Discussion.

Thank you for this suggestion. We have deleted the initial summary of study results (line 338-343) and trimmed content that is included in supplementary material (line 380-381). 

6. PLOS authors have the option to publish the peer review history of their article (what does this mean?). If published, this will include your full peer review and any attached files.

Do you want your identity to be public for this peer review? For information about this choice, including consent withdrawal, please see our Privacy Policy.

Reviewer #1: No

Reviewer #2: No

Reviewer #3: Yes: Anne Marie Bakke, Prof. Dr.med.vet.

---

## [Decision Letter · Decision Letter 1]

3 Nov 2023

PONE-D-23-18094R1Domestic dogs maintain clinical, nutritional, and hematological health outcomes when fed a commercial plant-based diet for a yearPLOS ONE

Dear Dr. Melgarejo,

Thank you for submitting your manuscript to PLOS ONE. After careful consideration, we feel that it has merit but does not fully meet PLOS ONE’s publication criteria as it currently stands. Therefore, we invite you to submit a revised version of the manuscript that addresses the points raised during the review process.

GIVEN THE SUGGESTION OF REJECTION FROM THE LAST REVIEWER I SOUGHT ADDITIONAL FEEDBACK FROM TWO REVIEWERS. I BELIEVE THE COMMENTS FROM REVIEWER #4 MUST BE ADDRESSED IN A REVISED VERSION BEFORE A FINAL DECISION CAN BE MADE.

We look forward to receiving your revised manuscript.

Kind regards,

Juan J Loor

Academic Editor

PLOS ONE

Journal Requirements:

Reviewers' comments:

Reviewer's Responses to Questions

**Comments to the Author**

1. If the authors have adequately addressed your comments raised in a previous round of review and you feel that this manuscript is now acceptable for publication, you may indicate that here to bypass the “Comments to the Author” section, enter your conflict of interest statement in the “Confidential to Editor” section, and submit your "Accept" recommendation.

Reviewer #1: All comments have been addressed

Reviewer #2: All comments have been addressed

Reviewer #4: (No Response)

2. Is the manuscript technically sound, and do the data support the conclusions?

Reviewer #1: Partly

Reviewer #2: Yes

Reviewer #4: Partly

3. Has the statistical analysis been performed appropriately and rigorously? 

Reviewer #1: I Don't Know

Reviewer #2: Yes

Reviewer #4: Yes

4. Have the authors made all data underlying the findings in their manuscript fully available?

Reviewer #1: Yes

Reviewer #2: Yes

Reviewer #4: Yes

5. Is the manuscript presented in an intelligible fashion and written in standard English?

Reviewer #1: Yes

Reviewer #2: Yes

Reviewer #4: Yes

6. Review Comments to the Author

Reviewer #1: (No Response)

Reviewer #2: The authors responded adequately to the questions presented and the limitations of the study were adequately highlighted, so I believe that the manuscript is suitable for publication.

Reviewer #4: While I agree that the paper and findings support the use of plant-based diets as the main source of nutrition, there is still the possibility of alternative nutrition being available and uncontrollable, as well as a lack of a control group to compare to. The use of the baseline as each individual control is quite far removed from the end of the study (a year later) with limited ability to understand aging or seasonal effects to the parameters. 1 - 11 years is a big range on top of potential breed confounding factors.

STUDY DESIGN:

L99-101: Why is there reference to a sample timeframe of every 3 months if it was disregarded for the analysis? If the samples were taken I feel it would be valuable to show the data, even if it is not analyzed in the final dataset.

RESULTS:

There is no mention of the questionnaire results. Due to their inclusion in the methods, this is a miss.

DISCUSSION:

This contains references to figures already described in the results (i.e. Fig 1, Fig 2 etc) and feels repetitive. I don't believe there is a need to re-reference the figure references.

L389: statement should include lack of analysis of Taurine in urine too.

L441-444: mention of questionnaire again but no clarity on any results/findings from this. This section implies that the questionnaire would give clarity or reassurance on the compliance to the study protocol (i.e. no meat-based treats), however the questions do not cover additional feeding explicitly. I am concerned that the questionnaire is inadequate to aid in ensuring compliance.

7. PLOS authors have the option to publish the peer review history of their article (what does this mean?). If published, this will include your full peer review and any attached files.

Reviewer #1: No

Reviewer #2: No

Reviewer #4: **Yes: **Jennifer Carolyn Coltherd

---

## [Author Response · Author response to Decision Letter 1]

16 Dec 2023

REBUTTAL LETTER

RESPONSE TO ACADEMIC EDITOR & REVIEWER COMMENTS

12/16/23

PONE-D-23-18094R1

Domestic dogs maintain clinical, nutritional, and hematological health outcomes when fed a commercial plant-based diet for a year

PLOS ONE

Response to Reviewer #4

Reviewer #4: While I agree that the paper and findings support the use of plant-based diets as the main source of nutrition, there is still the possibility of alternative nutrition being available and uncontrollable, as well as a lack of a control group to compare to. The use of the baseline as each individual control is quite far removed from the end of the study (a year later) with limited ability to understand aging or seasonal effects to the parameters. 1 - 11 years is a big range on top of potential breed confounding factors.

- We thank the reviewer for this important observation and agree that there are always inherent limitations of different study designs, including the one chosen for this study. The possibility of alternative nutrition is inevitable in a study that includes client-owned dogs living in households outside a restrictive laboratory environment. While one can focus on the limitations of work with client-owned dogs, there are also clear advantages. The study of laboratory beagles in a controlled environment does not reflect the true diversity of canine lifestyles and breeds within different home environments. We find the advantages of studying client-owned companion dogs far outweigh any concerns. 

Having dogs serve as their own controls is considered a statistically sound approach.

If we had introduced a separate control group, we would have added more variation. The study did not attempt to understand effects of aging or seasonal changes and this was never within the scope. All adult dogs who were volunteered for the study were enrolled. It is correct that one could have elected a narrow range of age, body weight, breed, etc. However, this would also have meant that we would have had a much more difficult time enrolling a sufficient number of dogs in this study. In other words, enrolling dogs in a clinical study is also subject to logistic limitations such as budget and location. 

STUDY DESIGN:

L99-101: Why is there reference to a sample timeframe of every 3 months if it was disregarded for the analysis? If the samples were taken, I feel it would be valuable to show the data, even if it is not analyzed in the final dataset.

- As mentioned in lines 94-98, all dogs were examined quarterly, however, samples were collected at 0, 6, and 12 months. Consequently, we show data from three timepoints. Also, analyzing samples quarterly would have required a significantly larger budget. 

RESULTS:

There is no mention of the questionnaire results. Due to their inclusion in the methods, this is a miss.

-We have added the following (lines 213-214) in response to the comment: “No adverse effects were reported for any of the dogs in response to the eight questions included in the client telephone questionnaire.”

DISCUSSION:

This contains references to figures already described in the results (i.e. Fig 1, Fig 2 etc.) and feels repetitive. I don't believe there is a need to re-reference the figure references.

-We appreciate the comment and understand that some readers might find this repetitive. Having said that, we have included the figure references to enhance readability for those who benefit from this approach and to improve accessibility. 

L389: statement should include lack of analysis of Taurine in urine too.

- Thank you for noting this omission. We have revised the text (line 379): “We did not measure taurine levels in whole blood or urine, which is a study limitation”

L441-444: mention of questionnaire again but no clarity on any results/findings from this. This section implies that the questionnaire would give clarity or reassurance on the compliance to the study protocol (i.e. no meat-based treats), however the questions do not cover additional feeding explicitly. I am concerned that the questionnaire is inadequate to aid in ensuring compliance.

-We have added the following text (line 433): “During the phone interviews, it was moreover confirmed that all canine participants were fed only the prescribed plant-based study diet/treats (i.e., no foods or treats containing animal-based products).”

---

## [Editor Report · Decision Letter 2]

2 Feb 2024

Domestic dogs maintain clinical, nutritional, and hematological health outcomes when fed a commercial plant-based diet for a year

PONE-D-23-18094R2

Dear Dr. Melgarejo,

We’re pleased to inform you that your manuscript has been judged scientifically suitable for publication and will be formally accepted for publication once it meets all outstanding technical requirements.

Kind regards,

Juan J Loor

Academic Editor

PLOS ONE
---

## [Editor Report · Acceptance letter]

20 Feb 2024

PONE-D-23-18094R2 

PLOS ONE

Dear Dr. Melgarejo, 

I'm pleased to inform you that your manuscript has been deemed suitable for publication in PLOS ONE. Congratulations! Your manuscript is now being handed over to our production team.

Kind regards, 

on behalf of

Dr. Juan J Loor 

Academic Editor

PLOS ONE